# Mechanical Performance of Bio-Based FRP-Confined Recycled Aggregate Concrete under Uniaxial Compression

**DOI:** 10.3390/ma14071778

**Published:** 2021-04-03

**Authors:** Elhem Ghorbel, Mariem Limaiem, George Wardeh

**Affiliations:** Laboratory of Mechanics and Materials of Civil Engineering (L2MGC), CY Cergy Paris University, 5 mail Gay LUSSAC, 95031 Neuville-sur-Oise—Cergy-Pontoise CEDEX, France; mariemlimaiem@gmail.com (M.L.); george.wardeh@cyu.fr (G.W.)

**Keywords:** concrete, recycled aggregates, bio-sourced composite, carbon epoxy composites

## Abstract

This research investigates the effectiveness of bio-sourced flax fiber-reinforced polymer in comparison with a traditional system based on carbon fiber-reinforced epoxy polymer in order to confine recycled aggregate concretes. The experimental investigation was conducted on two series of concrete including three mixtures with 30%, 50%, and 100% of recycled aggregates and a reference concrete made with natural aggregates. The concrete mixtures were intended for a frost environment where an air-entraining agent was added to the mixture of the second series to achieve 4% air content. The first part of the present work is experimental and aimed to characterize the compressive performance of confined materials. The results indicated that bio-sourced composites are efficient in strengthening recycled aggregates concrete, especially the air-entrained one. It was also found that the compressive strength and the strain enhancement obtained from FRP confinement are little affected by the replacement ratio. The second part was dedicated to the analytical modeling of mechanical properties and stress–strain curves under compression. With the most adequate ultimate strength and strain prediction relationships, the full behavior of FRP-confined concrete can be predicted using the model developed by Ghorbel et al. to account for the presence of recycled aggregates in concrete mixtures.

## 1. Introduction

Concrete structures are subjected to accidental loads such as seismic or mechanical shocks, excessive loads, temperature gradients, design flaws or defective implementation, which could lead to material damage and failure. Therefore, their strengthening is a possible issue and requires advanced research, and this is especially true if the aggregates come from construction and demolition wastes (CDWs) known as recycled concrete aggregates (RCA). Several studies have been conducted on recycled aggregates concretes (RAC) during these last decades, since the reuse of CDWs is defined as a priority area in the EU according to the Circular Economy Action Plan [1].

It is currently admitted that the incorporation of RCA leads to recycled aggregates concretes RAC with higher porosity and ductility and lower permeability, durability resistance, and mechanical properties than those of natural aggregates concrete [2,3,4,5,6,7,8,9,10,11]. However, the reuse of RCA to manufacture structural concrete is necessary to decrease the environmental impact of buildings and civil engineering works as well as to minimize the consumption of natural resources. Hence, the European Directive requirements stipulate a 70% recovery target for C&DW in 2020 and at least 25% reuse in the buildings and constructions field [12].

Many investigations have explored since 2012 the potential of confining RACs by fiber-reinforced polymers (FRPs) in order to enhance their compressive strength and to increase their ductility using both experimental and analytical approaches [13,14,15,16,17,18,19,20]. In all these studies, only coarse recycled aggregates were incorporated in concretes with replacement ratios varying from 20% to 100%. Moreover, the main findings indicate that both the compressive strength and the deformation of RAC are enhanced under FRP warps or tubes confinement by comparison to unconfined RACs.

Fiber-reinforced polymers (FRPs) became the most used materials for concrete structural elements repairing and strengthening since 1990 [21] because of their outstanding characteristics such as light weight, high tensile strength, and processability in addition to their corrosion resistance [22,23]. The engineering application of FRP composites has increased rapidly thanks to their availability in various flexible forms such as thin sheets, which can be warped around structural elements more easier than steel or concrete plates [22].

The most common composites used are based on epoxy resins systems reinforced by fibers such as carbon (CFRP), glass (GFRP), aramid (AFRP), and basalt (BFRP). Carbon fibers as well as aramid ones are manufactured respectively from polyacrylonitrile (PAN) and para-phenyleneterephthalamide (PPD-T), while glass fibers and basalt ones are obtained respectively from silica and igneous rocks. Their production is responsible for toxic gas emission impacting the environment. Hence, these fibers are not biodegradable and non-sustainable materials. The matrix is generally a thermosetting resin. The most used are epoxy, polyester, and vinyl ester resins. These polymers contain bisphenol A (BPA), which leads to health damages and are obtained from nonrenewable fossil resources.

The European directives enforce not to use materials produced from non-renewable resources, non-biodegradables, having unhealthy effects on humans and a high carbon footprint. Using RACs aims to diminish natural resources consumption and to achieve the European target for CDWs management. Confining RACS by conventional composites such as CFRP, GFRP, AFRP, or BFRP is not a sustainable solution and acts against European recommendations. Therefore, the use of bio-based composites for the reinforcement and repairing of concretes is recommended. Fewer studies have been carried out to investigate the efficiency of natural fibers reinforcing polymers, and the results are promising [24,25,26,27,28,29]. A recent study has focused on the use of bio-based epoxy matrix reinforced with flax fibers to reinforce and repair concrete [30], and the obtained results were encouraging and outline that bio-sourced confined C35/45 concretes have interesting performances.

This paper intends to demonstrate how far the bio-resourced composite can be developed based on the work of Limaiem et al. [30], replacing the CFRP composite for strengthening recycled aggregates concretes (RACS), as no published studies deal yet with this subject. Moreover, the target RACs of this research are formulated using both recycled sand and gravel, while to date, the FRP confinement is limited to RACs incorporating only coarse recycled aggregates.

Moreover, the target RACs of this research are formulated using both recycled sand and gravels, while the published research is limited to FRP-confined RACs incorporating only coarse recycled aggregates. Hence, four formulations have been studied differing in the replacement ratios: 0% (the reference), 30%, 50%, and 100%. In addition, the RACs were intended for a frost environment and should be resistant to frost exposure class (XF1–XF3). To hit this objective, the four mixtures were adjusted to contain 4% of entrained air. The confinement effectiveness by CFRP in comparison to bio-composites of RACs was experimentally studied, and the obtained stress–strain curves were modeled using an analytical approach. To this end, the main objectives of this work are:-The valorization of recycled sand in concrete mixtures without and with air entraining agent.-The formulation of concretes with a fully recycled granular skeleton.-An investigation on the effectiveness of unidirectional flax fiber-reinforced bio-sourced epoxy resin in confining air entrained recycled aggregates concrete.-A comparison between the effectiveness of bio-sourced FRP and the traditional CFRP.-A validation of the applicability of an analytical stress–strain model to confined air entrained recycled aggregates concrete with any type of fibers.

## 2. Materials and Methods

### 2.1. Concrete

The raw materials used to formulate concrete were: -Cement type CEM II/A-L 42.5N, Betocarb HP-OG,-Limestone fillers manufactured by OMYA SAS,-MC PowerFlow 3140 superplasticizer-Natural river sand called NS 0/4, two crushed natural gravels NG1 4/10 and NG2 6.3/20, and recycled aggregates provided from construction demolition wastes designed by RS0/4, RG1 4/10, and RG2 6.3/20.

Four mix proportions initially proposed within the framework of the ANR ECOREB project [31] were elaborated. The reference mixture is named C0R-0R and corresponds to a concrete designed to achieve a C35/45 compressive strength class and a consistence class of S4 with a target slump of 19 + 1 cm. Three mixtures were produced based on the reference one by incorporating recycled aggregates from CDW named C30R-30R, C0R-100R, and C100R-100R (Table 1). The nomenclature CxR-yR was defined as follows: x represents the replacement ratio of recycled sand (RS) by the total weight of sand and y represents the replacement ratio of recycled gravel (RG) by the total weight of gravels. In this work, a new by mass equivalent replacement ratio, named (Γ*_m_*), was defined by Equation (1).
(1)$$\Gamma_{m}=\frac{\left(M_{RS}+M_{RG}\right)}{\left(M_{NA}+M_{RA}\right)}$$
where *M_RS_* is the mass of RS in 1 m^3^ of concrete, *M_RG_*, *M_NA_*, and *M_RA_* are respectively, the mass of recycled gravels, the total mass of natural aggregates (NA), and the total mass of recycled aggregates RCA in 1 m^3^ for all mixtures. 

Compressive tests were conducted on cylindrical specimens (15 cm of diameter and 30 cm of length) at 90 days age, and the obtained properties for each concrete formulation are summarized in Table 2. It can be observed that at 90 days age, the mechanical properties are slightly affected by the introduction of recycled concrete aggregates (RCAs). As a matter of fact, a slight diminution is observed for compressive strength and elastic modulus (between 10% and 18% for *f_cm_* and 8% and 20% for *E_c_*), while an increase is noticed for the peak strain “*ε*_*c*1_” for non-entrained air formulation. However, it appears that the introduction of an air-entraining agent for a given formulation leads to the decrease in the mechanical properties.

### 2.2. Composites

Three types of composites were used in order to investigate their effectiveness in improving the mechanical behavior and durability of recycled aggregates concretes:

The first composite FOREVA TFC, called “CBF”, is a carbon bidirectional woven fabric with 0°/90° fiber orientation reinforced with epoxy resin commercialized by FREYSSINET.

The second one “SIKAWRAP-230C”, designed as “CUS”, is unidirectional carbon fiber fabric reinforced with epoxy polymer delivered by SIKA.

The last one named “FUB” is a unidirectional flax fiber fabric associated with EnviPOXY^®^530 product cross-linked with Phenalkamine NX5619. The development of the bio-sourced epoxy resin and its hardener was done in the framework of the ANR MICRO [32]. The composites have been made by contact molding. The mold (a plate 30 cm × 25 cm) was first treated with a release agent and covered with a plastic film before applying the required quantity of the resin and hardener. Then, the fibers’ fabrics (one ply for carbon fabric and two plies for flax ones) were placed and manually compacted to ensure that the fabric is fully permeated in the polymer. After 7 days of hardening under laboratory climatic conditions, the plate was cut into rectangular specimens of 25 × 30 cm devoted to the tensile test. Composite samples were subjected to uniaxial tensile tests in the two main directions 0° (fiber main direction) and 90°. The mechanical properties of the three composites are summarized in Table 3. Results show that the tensile properties of the bio-sourced composite are weaker than those of CFRP even if 2 fabric layers are used instead of one for carbon composites. Moreover, the highest properties for CBF in the 90° direction are due to the presence of fibers in this direction (Vf = 30%).

### 2.3. Specimen Preparation and Compressive Test

Cylindrical specimens with dimensions 15 × 30 cm were casted and stored in water at room temperature for 90 days. After wiping with a wet cloth, the resin was applied on all the side surface of the sample with a paintbrush, as shown in Figure 1. The fabric was applied delicately on the side surface, avoiding air pockets by using a roller. Only one ply was used for both carbon composites and two plies for flax fiber composite with an overlap length of 10 cm to ensure the total confinement of the specimens. The reinforced samples were stocked under ambient temperature for four days before the test. 

In order to examine the effect of FRP on the concrete behavior, cylindrical specimens were tested under monotonic compressive loading. Tests were conducted using a hydraulic press INSTRON SCHENCK with a capacity of 3000 kN by imposing a displacement rate of 1 mm/min. Axial strains in the middle portion of the specimens were measured using 3 LVDTs spaced 120° along the circumference of a crown. 

Splitting tensile and fracture tests were not carried out, although these properties are essential for the evaluation of the overall behavior of concrete structures. However, these properties can be predicted through the compressive characteristics as shown by Sucharda et al. [33]. 

## 3. Results

### 3.1. RAC without Air Entrainment Agent

Figure 2 shows an example of the concrete confining effect with polymer CFRP. Such treatment provides a great ductility to concrete with brittle fracture. Three different phases in the behavior of confined concrete are observed. The first part of the curve corresponds to the elastic behavior of the concrete. The second part is the transition zone where the stress–strain of the concrete starts to soften with the formation and the propagation of cracks leading to the dilation of concrete that bears against the composite jacket to activate it. The third portion of the curve is linear and ascending due to the full activation of the FRP. This part indicates that confinement is high to moderate. The confining stress increases up to the sudden brittle failure of the FRP jacket produced by fibers breaking.

Figure 2b illustrates an example for experimental curves obtained under compression for the reference concrete. The obtained compressive stress–strain curves indicate that confinement is high for all the used composite systems. The difference is observed for specimens confined with flax composites in the transition stage of the curve. In fact, it seems that the concrete dilation required to fully activate the FUB jacket was more important than in the case of CBF and CUS.

Using CFRP to strengthen recycled aggregates concrete is less efficient as the ratio of recycled aggregates increases (Figure 3). It should be noticed that the compressive strengths of confined recycled aggregates concrete are at least restored or are higher than those of unconfined reference concrete. Hence, strengthening RAC by CFRP can be useful. The use of FUB allows reinforcing recycled aggregates concrete but not to restore the strength, so it can reach this of the unconfined reference. Regarding the ductility, confinement leads to improve it in the same manner whatever the type of composite used for strengthening recycled aggregates concrete. Even though the flax composite has a clearly lower effect on strengthening RAC than the carbon one, it is still a very interesting material according to the enhanced compressive strength and ductility.

The failure mode of samples reinforced with three types of FRP is shown in Figure 4. All confined specimens failed in a sudden and explosive manner with the rupture of FRP jackets due to the expansion of core concrete. The failure mode had no correlation with to the type and the thickness of the used FRP and to the replacement ratio of recycled aggregates. Inside the FRP sheets, the concrete core was fully crushed. 

### 3.2. Strengthening Entrained Air RAC

The general stress–strain curves shown in Figure 5 indicate that the efficiency of confining air-entrained recycled aggregates by the FRP jacket. However, FUB leads to low/moderate confinement with a descending curve, while CUS and CBF generate high confinement with an ascending curve. 

The strengthening of air-entrained concrete is efficient regarding the compressive strength as well as ductility. CFRP is more efficacious than the bio-sourced composite “FUB”, as illustrated in Figure 6. However, the ability of FUB to strengthen air-entrained concrete and to increase its ductility becomes comparable to that of CFRP when the incorporation ratio of recycled aggregates exceeds 30%. The CBF composite leads to the best results because the fabric is bidirectional and not unidirectional as is the case of CUS and FUB.

### 3.3. Analytical Modeling

Figure 7 represents the confinement ratio (the ratio of the hoop confining pressure *f_l_* of FRP to the compressive strength *f_cm_* of unconfined concrete) as a function of the equivalent replacement ratio Γ*_m_* for both series of concrete. Concerning the confinement pressure, *f_l_*, it can be given by relation 2 as a function of the mechanical properties of the used FRP, the thickness, and the number of FRP layers as well as the diameter of the tested specimen:(2)$$f_{l}=2n\left(\frac{t}{D}\right)f_{t}$$
where *n*, *t*, and *f_t_* are the number of FRP layers, the thickness of the FRP jacket, and the tensile strength, respectively, while D is the specimen’s diameter. It can be observed from Figure 5 that the used FRP imposes a different confining pressure, which essentially depends on the characteristics of the fibers used. CBF is the most efficient fiber since it has the highest tensile strength accompanied with the highest thickness followed by CUS and finally FUB, which has the lowest resistance. It is worth noting that CBF and CUS were used in one layer, while bio-based was used in two layers. The comparison between Figure 5a,b shows that for the same type of FRP and the same replacement ratio, the confining ratio is more important when an air-entraining agent is used.

Figure 8 shows the variation of the compressive strength gain ratio (*f_cc_/f_cm_*) against the replacement ratio for both series of concrete. It can be observed that the gain decreases with the replacement ratio for the series CxR-yR, while it increases for the series CxR-yR-4 containing an air-entraining agent, especially with CBF fiber. It can be also shown that the compressive strength enhancement obtained from FRP confinement is little affected by the replacement ratio. The same statement was found by Chen et al. [18]. The strain gain ratio (the ultimate strain of confined concrete, *ε_cc_*, to the peak strain of unconfined concrete named *ε_c_*_1_) follows the same variation of the strength gain with the variation of the replacement ratio, as shown in Figure 9. However, these findings do not agree with the conclusions of Zhou et al. [15], who found that both the strength gain ratio and strain gain ratio were basically the same regardless of the change in the replacement ratio. The effectiveness is better for the series CxRyR-4 and according to Choudhury et al. [16], this phenomena can be attributed to the high dilatation ability of concrete containing air voids under axial compression where the confining pressure rapidly increases as a result of the increase in the lateral dilatation. For a higher confining pressure, the enhancement of the compressive strength is higher. All of these results show that the use of recycled concrete aggregates in a harsh environment, where the use of an air-entraining agent is necessary, is quite possible for elements under compressive loads when this concrete in confined by FRP whatever its nature if its properties are sufficient to create a confining pressure able to prevent lateral deformation. Conversely, in the frost zone, confining non air-entrained concrete can prevent water from moving and escaping from the surface, which could consequently increase the magnitude of the hydraulic pressure and ice crystallization pressure [34] and increase concrete damage [35].

The stress recovery is defined as the ratio between the ultimate strength of confined concrete, *f_cc_*, and the compressive strength of NAC for each series named *f_cm,NAC_*. The variation of this ratio against the replacement ratio is illustrated in Figure 10 where it can be observed that CBF and CUS fibers allowed a total strength recovery that has been lost due to the incorporation of recycled aggregates. Concerning a bio-based fiber, FUB, it led to the recovery of the strength of the air-entraining concrete with an increase of 8% for the mixture C100R-100R-4. It should be noted that a single layer of CUS and two layers of FUB fibers allowed the mixture C100R-100R to restore, respectively, 100% and 91% of the NAC’s compressive strength. This finding is too encouraging knowing that authors used two or three layers to strengthen the tested specimens [15,18,36].

Figure 11a represents the relationship between the stress gain ratio (*f_cc_/f_cm_*) as a function of the confining ratio (*f_l_/_fcm)_*, while Figure 11b illustrates the relationship between the strain gain ratio (*ε_cc_/ε_c_*_1_) against the confining pressure. It can be shown that the stress gain ratio and the strain gain ratio have a linear growth trend with the confining pressure regardless the type of the used FRP, the RCA replacement ratio, as well as the entrained air content. These trends, which are in perfect agreement with the results of the literature [15,18,36,37], confirm that the mechanical properties of FRP-confined RAC significantly depend on the confinement ratio, as is the case for the FRP-confined NAC. 

The strength and strain enhancement ratios were calculated using analytical relationships presented in Table 4 and Table 5, and the performance of each model was evaluated using the correlation coefficient *R*^2^ given by Equation (3). A value of *R*^2^ close to one indicates that the experimental points approach the model, while a value near zero means that the experimental points are too scattered around the line describing the model.
(3)$$R^{2}=1-\frac{SSE}{SST}=1-\frac{\sum_{i=1}^{n}{\left(y_{i}-{\overset{⌢}{y}}_{i}\right)}^{2}}{\sum_{i=1}^{n}{\left(y_{i}-\overset{⌢}{y}\right)}^{2}}$$
with
*SSE*: the residual sum of squares*SST*: the total sum of squares*n*: the number of experimental points*y_i_*: the *i*th experimental measurement $\hat{y_{i}}$: the predicted value$\hat{y}$: the mean value. 

The obtained results in terms of R² show that the model of Lam and Teng [38] (Equation (4)) is the most suitable for the prediction of the stress gain, while the model of Jiang and Teng [39] (Equation (11)) is the most suitable for the strain gain prediction. The experimental versus predicted results are depicted in Figure 12. 

The model given by Equation (13) was adopted to describe the full stress–strain behavior of both unconfined and FRP-confined NAC and RAC. This model proposed initially by Popovic and adopted later by Mander [42] was modified in the work of Ghorbel et al. [43] to account for the presence of RCA for unconfined concrete. It requires only the knowledge of the compressive strength (*f_cm_* of *f_cc_*); then, the tangent elastic modulus can be calculated based on the mean compressive strength and the replacement ratio according to Equation (14). The peak strain for unconfined concrete can be calculated according to Equation (15), where it was found that this strain depends only on the compressive strength regardless of the replacement ratio [3,43].
(13)$$\frac{\sigma }{f_{cc}}=\frac{\beta_{m}\left(\frac{\varepsilon }{\varepsilon_{cc}}\right)}{\beta_{m}-1+{\left(\frac{\varepsilon }{\varepsilon_{cc}}\right)}^{\beta_{m}}}$$
with $\beta_{m}=\frac{1}{1-\frac{f_{cc}}{E_{ci}\varepsilon_{cc}}},$$\beta_{m}=\frac{1}{1-\frac{f_{cc}}{E_{ci}\varepsilon_{cc}}}$, $E_{ci}=\frac{E_{c}}{0.85}$
$E_{ci}=\frac{E_{c}}{0.85}$ where *E_ci_* is initial elastic modulus.
(14)$$E_{c}=17553\left(1-0.131\Gamma_{m}\right){\left(\frac{f_{cm}}{10}\right)}^{0.42}$$
(15)εc1(000)=1.1(fcm)0.175

In Figure 13 and Figure 14a, a comparison is shown between the experimental and full stress–strain curves for both series of concrete without and with air-entraining agent. Based on the compressive strength of unconfined concrete, the stress enhancement was calculated according to Equation (4), while the strain gain was calculated according to Equation (11). It can be observed that the model predicts quite well the overall behavior whatever the aggregates and the FRP used. It is worth mentioning that the model is sensitive to the value of *β_m_*. Indeed, when the experimental peak strain of unconfined concrete is used instead of peak strain calculated using Equation (15), the agreement between analytical and experimental curves is better.

## 4. Conclusions

In this paper, the axial behavior of FRP-confined recycled aggregate concrete was investigated. Two series of concrete without and with 4% air-entraining agent and various replacement ratios of recycled aggregates were prepared and tested to examine the performance of three types of FRP. The used FRP were a carbon bidirectional woven fabric with 0°/90° fiber orientation reinforced with epoxy resin named CBF, a unidirectional carbon fiber fabric reinforced with epoxy polymer (CUS), and a unidirectional flax fiber fabric associated with EnviPOXY^®^530 product called FUB. The obtained experimental results lead to the following concluding remarks:-Confining recycled aggregate concrete by the unidirectional flax fibers reinforced bio-sourced epoxy resin, FUB, is significant for 4% air-entrained recycled aggregates concrete, and its effectiveness is comparable to the effectiveness of traditional composites based on carbon fibers CBF and CUS.-For the same type of FRP and the same replacement ratio, the confining ratio is more important when an air-entraining agent is used.-The compressive strength and the strain enhancement obtained from FRP confinement are little affected by the replacement ratio.-Bio-based fiber, FUB, led to the recovery of the strength of the air-entrained concrete with an increase of 8% for the mixture C100R-100R-4. Moreover, for C100R-100R, a single layer of CUS and two layers of FUB fibers allowed restoring 100% and 91% of the NAC’s compressive strength.-The performance of the model developed by Ghorbel et al. is satisfactory to predict the full stress–strain curves for both series of studied concrete in unconfined and confined configurations.

The experimental findings of this paper are mainly based on the compressive behavior of laboratory-size specimens. For structural reinforced concrete elements, more research is needed to validate the confinement effectiveness of bio-based fibers for air-entrained recycled aggregates concrete in a harsh environment. One might think also that confining and strengthening would be also benefits for rebar preservation against the ingress of chemical substances. 

## Figures and Tables

**Figure 1 materials-14-01778-f001:**
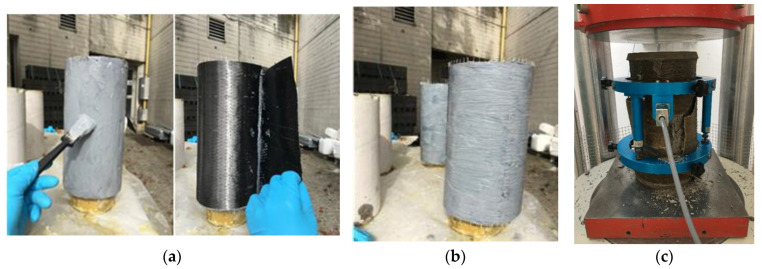
Preparing specimens with CUS jackets. CUS: unidirectional carbon fiber fabric reinforced with epoxy polymer. (**a**) resin application then CUS fabric laying, (**b**) manual dry impregnation of fabric, (**c**) test setup and instrumentation.

**Figure 2 materials-14-01778-f002:**
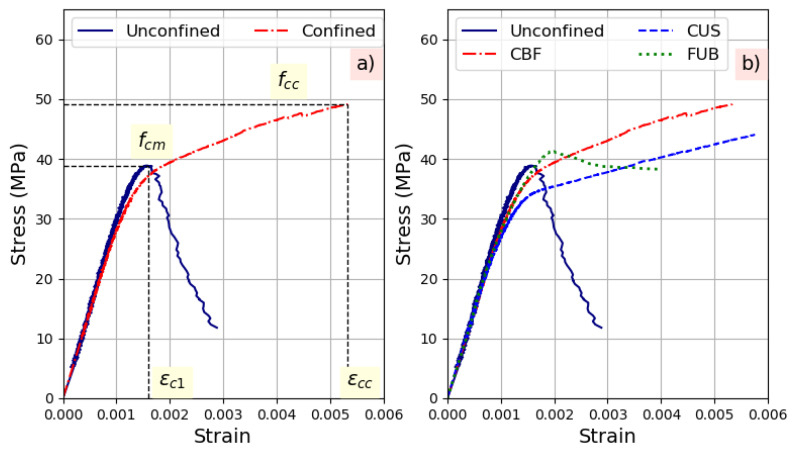
Behavior under compression tests of strengthened reference concrete C30R-30R: (**a**) Schematic stress–strain curve and main parameters, (**b**) Experimental compressive behavior.

**Figure 3 materials-14-01778-f003:**
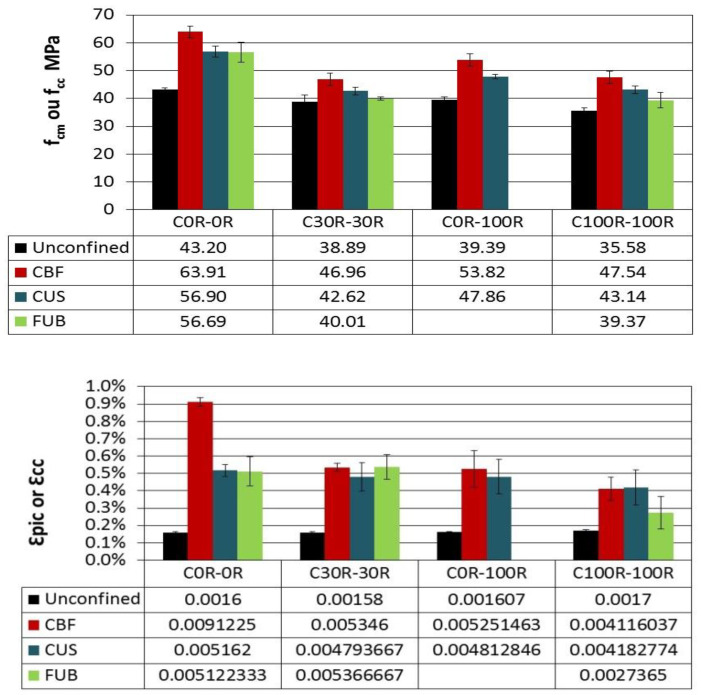
Ultimate strength *f_cc_* and ultimate confined concrete strain *ε_cc_* of recycled aggregates concretes (RAC) confined by fiber-reinforced polymers (FRP) composites.

**Figure 4 materials-14-01778-f004:**
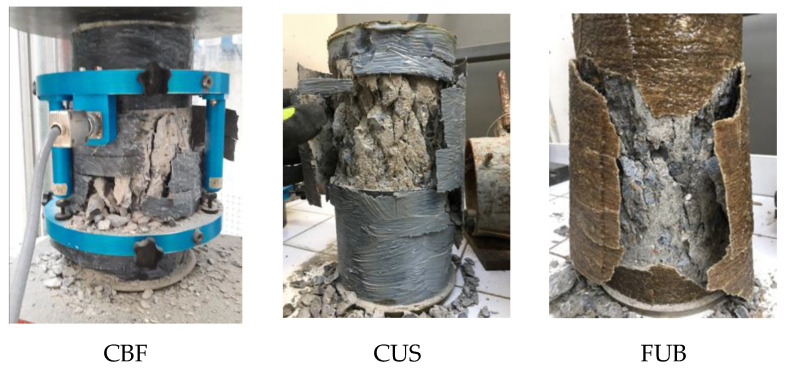
Failure mode of specimens.

**Figure 5 materials-14-01778-f005:**
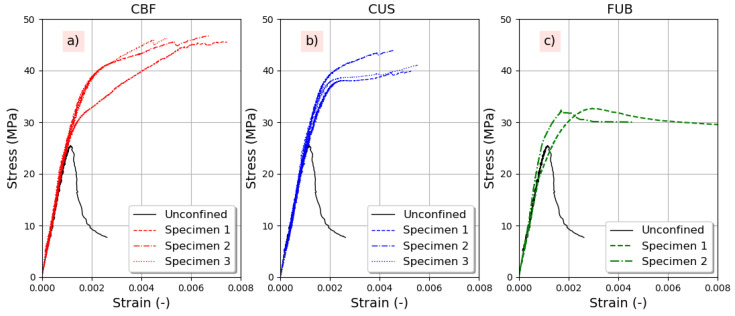
Compressive experimental curves of FRP-reinforced RAC-100R-100R-4.

**Figure 6 materials-14-01778-f006:**
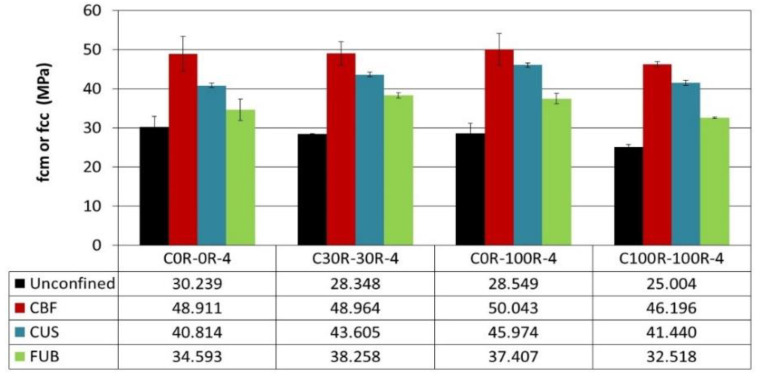
Ultimate strength *f_cc_* and ultimate confined concrete strain *ε_cc_* of air-entrained RAC confined by FRP composites.

**Figure 7 materials-14-01778-f007:**
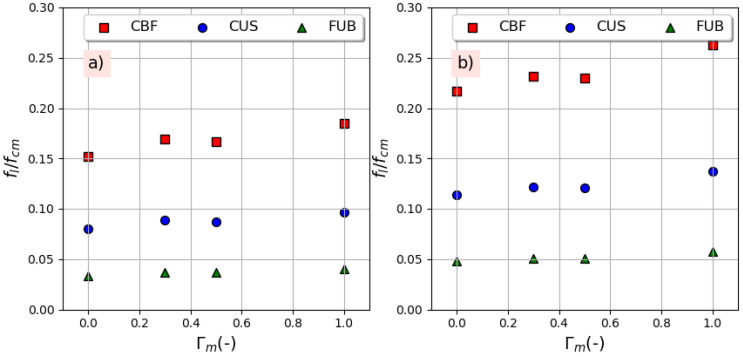
Confinement effectiveness with replacement ratio: (**a**) without air-entraining agent, (**b**) with 4% air-entraining agent.

**Figure 8 materials-14-01778-f008:**
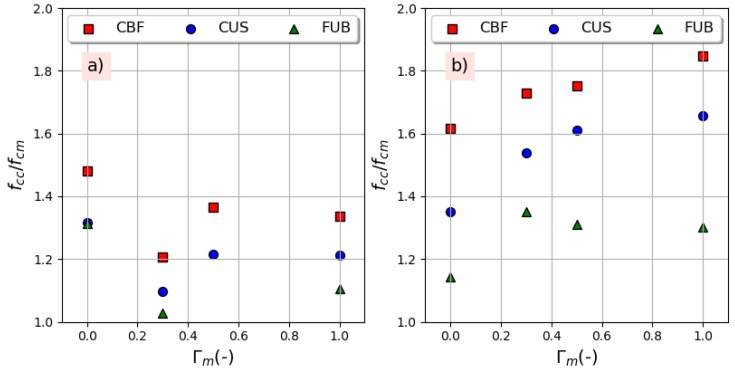
Strength enhancement with replacement ratio: (**a**) without air-entraining agent, (**b**) with 4% air-entraining agent.

**Figure 9 materials-14-01778-f009:**
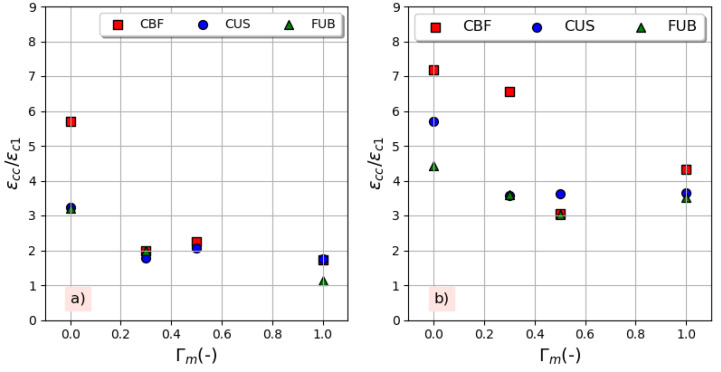
Strain efficiency with replacement ratio: (**a**) without air-entraining agent, (**b**) with 4% air-entraining agent.

**Figure 10 materials-14-01778-f010:**
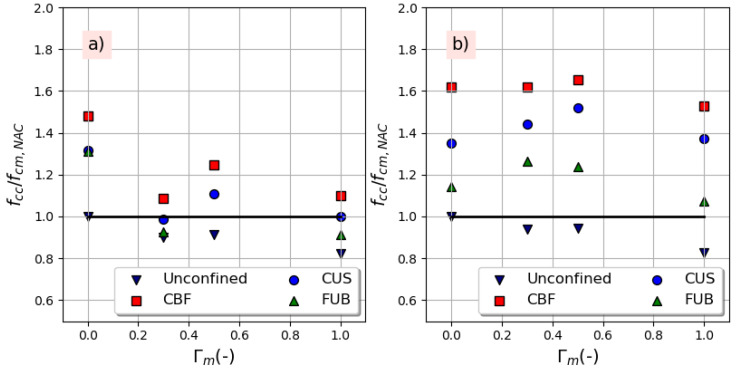
Strength recovery with replacement ratio: (**a**) without air-entraining agent, (**b**) with 4% air-entraining agent.

**Figure 11 materials-14-01778-f011:**
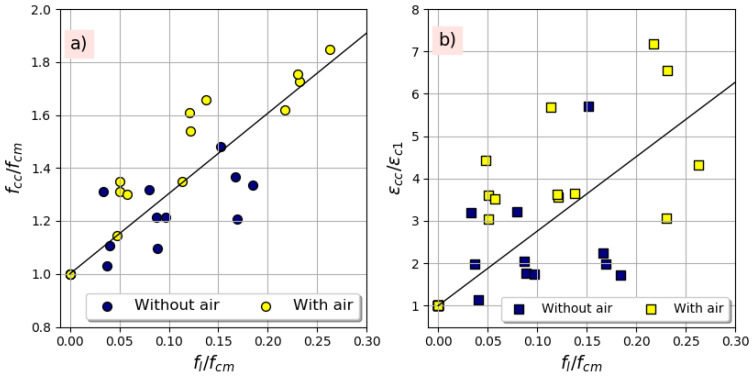
Effect of confinement ratio on (**a**) strength gain ratio, (**b**) strain gain ratio.

**Figure 12 materials-14-01778-f012:**
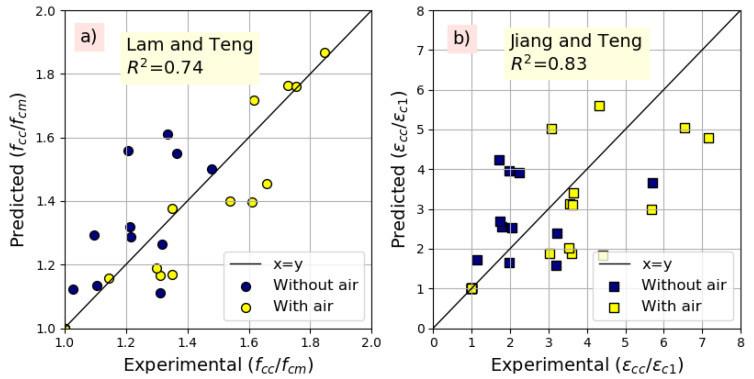
Predicted and experimented: (**a**) ultimate strength, (**b**) ultimate strain.

**Figure 13 materials-14-01778-f013:**
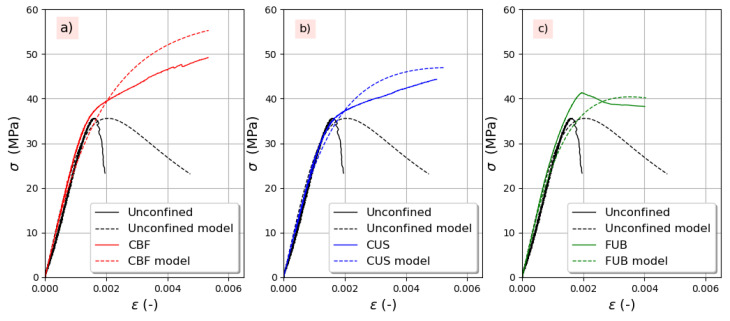
Experimental versus predicted stress–strain curves for concrete without an air-entraining agent: (**a**) CBF, (**b**) CUS, (**c**) FUB.

**Figure 14 materials-14-01778-f014:**
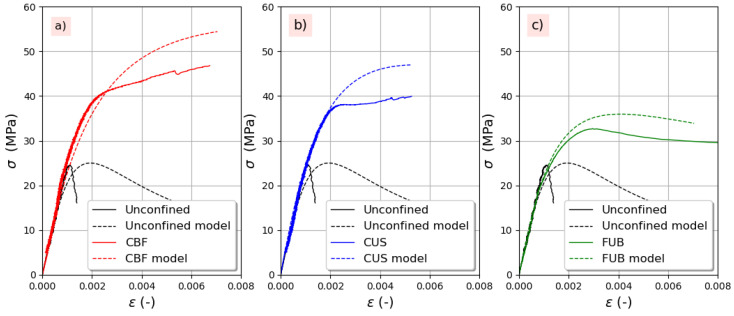
Experimental versus predicted stress–strain curves for concrete with 4% air-entraining agent: (**a**) CBF, (**b**) CUS, (**c**) FUB.4. Conclusions

**Table 1 materials-14-01778-t001:** Mix proportions of concrete mixtures.

Content (kg/m^3^)	Without Air-Entraining Agent				With Air-Entraining Agent			
	C0R-0R	C30R-30R	C0R-100R	C100R-100R	C0R-0R-4	C30R-30R-4	C0R-100R-4	C100R-100R-4
$\Gamma_{m}$ (%)	0	30	50	100	0	30	50	100
Total	185	220	238	284	178	171	175	163
Cement	299	321	336	381	287	299	310	339
Limestone filler	58	44	53	70	56	41	49	62
NS (0/4)	771	491	782	0	740	457	721	0
RS (0/4)	0	235.2	0	728.6	0	199	0	591
NG1 (4/10)	264	168	0	0	254	157	0	0
RG1 (4/10)	0	151.4	168.4	318.7	0	132	146	266
NG2 (6.3/20)	810	542	0	0	778	505	0	0
RG2 (10/20)	0	175.2	728.4	464.6	0	153	628	388
Superplasticizer	2.1	1.64	2.18	2.78	2	1.5	2	2.5
Air-Entraining agent	0	0	0	0	2.6	2	1.6	1.5
**Fresh properties**								
Slump (cm)	20	20	20	19	18	19	18	20
Air content (%)	1.8 ± 0.2	2.1 ± 0.1	2.4 ± 0.1	2.9 ± 0.1	4 ± 0.1	4 ± 0.1	4 ± 0.2	4 ± 0.2
	≈2.5				≈4			

**Table 2 materials-14-01778-t002:** Mechanical properties of the tested formulations at 90 days age.

Properties	Without Air-Entraining Agent				With Air-Entraining Agent			
	C0R-0R	C30R-30R	C0R-100R	C100R-100R	C0R-0R-4	C30R-30R-4	C0R-100R-4	C100R-100R-4
*f_cm_* (MPa)	43.20	38.89	39.39	35.58	30.239	28.348	28.549	25.004
*E_c_* (GPa)	32.21	29.75	26.08	26.03	29.540	28.806	26.424	24.536
*ε* _*c*1_	0.00160	0.00197	0.0021	0.0022	0.0013	0.00147	0.0013	0.0013

**Table 3 materials-14-01778-t003:** Composites tensile properties.

Fiber Direction	Composite	Thickness (mm)	f_t_ (MPa)		E (MPa)		ε_ultimate_	
			Average	SD	Average	SD	Average	SD
0°	CBF	0.48	1026.5	75.16	64,300	9300	0.018	0.005
	CUS	0.129	2001	156.24	114,000	54,300	0.0212	0.008
	FUB	0.25	216.29	83.14	27,000	2670	0.01	0.0044
90°	CBF	-	938.75	100.2	27,600	3260	0.035	0.004
	CUS		38.347	12.28	23,100	6060	0.002	0.0005
	FUB		86.18	11.12	15,110	760	0.006	0.0009

**Table 4 materials-14-01778-t004:** Ultimate strength models.

N°	Reference	Strength Model	Equation Number	*R* ^2^
1	Lam and Teng [38]	$\frac{f_{cc}}{f_{cm}}=1+3.3\left(\frac{f_{l}}{f_{cm}}\right)$	(4)	0.74
2	Bisby et al. [36]	$\frac{f_{cc}}{f_{cm}}=1+2.425\left(\frac{f_{l}}{f_{cm}}\right)$	(5)	0.68
3	Tamuzs et al. [37]	$\frac{f_{cc}}{f_{cm}}=1+4.2\left(\frac{f_{l}}{f_{cm}}\right)$	(6)	0.46
4	Youssef et al. [40]	$\frac{f_{cc}}{f_{cm}}=1+2.25{\left(\frac{f_{l}}{f_{cm}}\right)}^{1.25}$	(7)	0.20
5	Jiang and Teng [39]	$\frac{f_{cc}}{f_{cm}}=1+3.5\left(\frac{f_{l}}{f_{cm}}\right)$	(8)	0.71
6	Nistico and Monti [41]	$\frac{f_{cc}}{f_{cm}}=1+2.09\left(\frac{f_{l}}{f_{cm}}\right)$	(9)	0.56
7	Mander et al. [42]	$\frac{f_{cc}}{f_{cm}}=-1.2541+2.254{\left[1+7.94\left(\frac{f_{l}}{f_{cm}}\right)\right]}^{0.5}-2\left(\frac{f_{l}}{f_{cm}}\right)$	(10)	-

**Table 5 materials-14-01778-t005:** Ultimate strain models.

N°	Reference	Strain Model	Equation Number	*R* ^2^
1	Jiang and Teng [39]	$\frac{\varepsilon_{cc}}{\varepsilon_{c1}}=1.0+17.5\left(\frac{f_{l}}{f_{cm}}\right)$	(11)	0.82
2	Mander [42]	$\frac{\varepsilon_{cc}}{\varepsilon_{c1}}=1.0+5.0\left[\left(\frac{f_{cc}}{f_{cm}}\right)-1\right]$	(12)	0.77

## Data Availability

The data presented in this study are available on request from the corresponding author.

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
