# Peer review of "Mechanical Performance of Bio-Based FRP-Confined Recycled Aggregate Concrete under Uniaxial Compression"

_materials, 2021, doi:10.3390/ma14071778_

Round 1
Reviewer 1 Report
The paper describes the experimental tests of FRP-confined recycled aggregate concrete, with various replacement ratios and different types of fibers, under uniaxial compression.
The results of the experimental campaign show that the compressive strength and the strain enhancement obtained from FRP confinement are little affected by the replacement ratio and that different types and numbers of layers can be used to restore NAC’s compressive strength.
It is suggested the insertion of some pictures of the specimens before and after the compression tests.
Author Response
The authors would like to thank the reviewer 1 for his constructive remarks which led to improve the quality of the manuscript. The detailed answers are given below and the corrections are made in blue in the revised text. All answers are given in the attached file

Reviewer 2 Report
The originality and the scientific value of the subject research are good.
One of the weaknesses of the manuscript is that the experimental program focuses on only one mechanical property.
The research area is Mechanical performance of bio-based FRP-confined recycled aggregate concrete.
The topic and experimental program are interesting, but the overall informational value of the research could be greater.
There is also a need to improve the processing of the manuscript and the use of the template.
The table is on two pages. (Table 1)
Why is select text red? (Table 2)
Provide abbreviations in the text and tables. eg SD
The text belongs to the next page (3. Results)
Why are there blank lines? -lines 171-181
The test is on the wrong side (Figure 4).
Why are there blank lines? -lines 308-311
The test is on the wrong side (Figure 8).
Text of Figures is small - larger Figures or edit text - the whole manuscript.
Complete the manuscript with pictures of experiments/composites and testing.
Other mechanical properties are also important for design use and further research. Concrete is a quasi-brittle material. It is important to be interested in modulus of elasticity, tensile strength, fracture energy and many more.
For a comprehensive solution to the problem, it would also be appropriate to solve the problem by numerical modelling.
Also, a state in the manuscript the exact test scheme and the method of measuring deformations and strain.
In the solved area of mechanical properties of concrete and recycled aggregate concrete and there is extensive research, which must be stated in the context of the presented research. The introduction part needs to be reworked and supplemented with appropriate information.
Sucharda, O.; Mateckova, P.; Bilek, V. Non-Linear Analysis of an RC Beam Without Shear Reinforcement with a Sensitivity Study of the Material Properties of Concrete. Slovak Journal Of Civil Engineering, 2020, 28(1), 33-43.
Makul, N.; Fediuk, R.; Amran, M.; Zeyad, A.M.; Murali, G.; Vatin, N.; Klyuev, S.; Ozbakkaloglu, T.; Vasilev, Y. Use of Recycled Concrete Aggregates in Production of Green Cement-Based Concrete Composites: A Review. Crystals 2021, 11, 232.
and many more.
Overall, it is necessary to improve the graphic level of the manuscript and check the document template.
There is a lack of criticism throughout the analysis of the information in the manuscript.
The part of the discussion/conclusion needs to be reworked and more presented on new knowledge of research in the context of current research.
In the opinion opponent, It is not possible to publish the manuscript in its current form. The manuscript must be revised.
Author Response
The authors would like to thank the reviewer 2 for his constructive remarks which led to improve the quality of the manuscript. The detailed answers are given below and the corrections are made in blue in the revised text. All answers are given in the attached file

Reviewer 3 Report
Comments
This paper studied the effectiveness of bio-sourced Flax Fibre reinforced polymer under compression. The outcome is interesting for readers. However, there are several aspects that need to be improved. The reviewer can only recommend for publication if the author satisfactorily address the following comments in the revised version.
- Can the author provide a test setup photo for uniaxial compression?
- Any scientific reason behind the selection of 30%, 50% and 100% of recycled aggregates?
- The failure mechanism of the specimen should be discussed more clearly.
- The novelty of the study should be highlighted at the end of introduction section. How this study is different from the published study in literature?
- How the outcome of this study will benefit researchers and end users? This need to be highlighted in introduction or end of conclusion.
- The background study on the confinement effect is insufficient. Recently, the confinement effect was studied in FRP composite jackets [Ref: State-of-the-art of prefabricated FRP composite jackets for structural repair] and railway sleeper application [Ref: Static behaviour of glass fibre reinforced novel composite sleepers for mainline railway track]. Suggest to include them in introduction section with proper citations to improve the background study.
I would be happy to see the revised version to understand how these comments are being addressed.
Author Response
The authors would like to thank the reviewer 3 for his constructive remarks which led to improve the quality of the manuscript. The detailed answers are given below and the corrections are made in blue in the revised text. All answers are given in the attached file

Round 2
Reviewer 2 Report
Thank you for the adjustments made.
The changes made the improvement of the manuscript.
The research area and results are from the context of the manuscript can better understand.
The manuscript contains all the main information.
The manuscript can be published in the journal.